# BSA-Seq for the Identification of Major Genes for EPN in Rice

**DOI:** 10.3390/ijms241914838

**Published:** 2023-10-02

**Authors:** Shen Shen, Shanbin Xu, Mengge Wang, Tianze Ma, Ning Chen, Jingguo Wang, Hongliang Zheng, Luomiao Yang, Detang Zou, Wei Xin, Hualong Liu

**Affiliations:** Key Laboratory of Germplasm Enhancement and Physiology & Ecology of Food Crop in Cold Region, Ministry of Education/College of Agriculture, Northeast Agricultural University, Harbin 150030, China; m15581871670@163.com (S.S.); 18713599521@126.com (S.X.); wmenggew@163.com (M.W.); 1784309455@163.com (T.M.); neau_chenning@163.com (N.C.); wangjg@neau.edu.cn (J.W.); zhenghongliang008@163.com (H.Z.); yaochang616@163.com (L.Y.); zoudtneau@126.com (D.Z.)

**Keywords:** *Oryza sativa* L., EPN, BSA-seq, haplotype, candidate genes

## Abstract

Improving rice yield is one of the most important food issues internationally. It is an undeniable goal of rice breeding, and the effective panicle number (EPN) is a key factor determining rice yield. Increasing the EPN in rice is a major way to increase rice yield. Currently, the main quantitative trait locus (QTL) for EPN in rice is limited, and there is also limited research on the gene for EPN in rice. Therefore, the excavation and analysis of major genes related to EPN in rice is of great significance for molecular breeding and yield improvement. This study used japonica rice varieties Dongfu 114 and Longyang 11 to construct an F_5_ population consisting of 309 individual plants. Two extreme phenotypic pools were constructed by identifying the EPN of the population, and QTL-seq analysis was performed to obtain three main effective QTL intervals for EPN. This analysis also helped to screen out 34 candidate genes. Then, EPN time expression pattern analysis was performed on these 34 genes to screen out six candidate genes with higher expression levels. Using a 3K database to perform haplotype analysis on these six genes, we selected haplotypes with significant differences in EPN. Finally, five candidate genes related to EPN were obtained.

## 1. Introduction

Rice is a major food crop, and increasing its yield is essential for ensuring food security. With the rapid development of the economy and society, the cultivated area is constantly decreasing, while the population is constantly increasing. Therefore, the improvement of rice yield has become an urgent problem to be solved in the future [1]. The improvement of the total rice yield largely depends on the yield per unit area, which is primarily determined by the factors that make up the yield composition [2]. The EPN, grains per panicle, and thousand grain weight are the three components of rice yield composition [3]. The composition factors of rice yield are interrelated and mutually constrained, all controlled by multiple genes, and are sensitive to the environment. The EPN per plant in rice is influenced by the number of tillers, while the number of tillers in rice is constrained by both fertilizer and genetic background. For example, treating nitrogen deficiency during the rice tillering stage and compensating with nitrogen fertilizer during the young panicle differentiation stage can significantly increase the number of effective panicles per plant [4]. The EPN of rice is not only easily affected by environmental factors but also a complex quantitative trait controlled by multiple genes. EPN can seriously affect rice yield, and having stable EPN is one of the most important characteristics of an ideal plant structure [5]. Therefore, improving the EPN of rice plants is the primary objective of rice breeding. The exploration and analysis of major genes related to EPN in rice, as well as the identification of their dominant haplotypes, are crucial for rice breeding and increasing rice yield.

The mapping and gene cloning of quantitative trait loci (QTLs) for important agronomic traits in rice, such as yield components, are the foundation and prerequisite for molecular breeding of high-yield and high-quality rice. QTL mapping is the process of determining the position of a QTL on a chromosome by analyzing the relationship between all DNA markers in the genome and the measurements of the quantitative trait phenotype. It often includes the steps of constructing a genetic linkage map, obtaining mapping populations, detecting the genotype and phenotype measurements of individuals in the segregating generation population, and analyzing the relationship between genotype and phenotype. Traditional QTL localization typically requires the use of molecular markers distributed throughout the entire genome to genotype many individuals in a population [6]. To ensure sufficient statistical power, this method requires genotype and phenotype analysis of a large number of offspring, as opposed to bulked segregant analysis (BSA), which only requires genotyping of individuals with extreme phenotypes [7]. BSA is used for gene mapping of qualitative traits or quantitative traits with major genes [8]. When used for quantitative trait mapping, this technique is also known as QTL-seq [9] and has wide applications in genetic breeding. In this method, two parents with significant differences in traits are selected for hybridization to obtain an offspring population with separated traits. From this population, a certain number of individuals with extreme phenotypes are selected for DNA mixed-pool sequencing. The differences in genotype frequency of each locus between the two mixed pools at the whole genome level are then calculated to determine the regions where QTLs related to traits are located. Initially, the BSA method was widely used to identify QTLs related to specific traits, such as disease resistance, color, and fertility. Now, it has been applied to QTLs and gene mapping of various levels and traits. Therefore, BSA-seq, used in this study to locate the main QTLs and explore the target gene, is the correct choice.

A significant number of QTLs controlling rice yield traits have been successfully mapped and cloned in genetic mechanism research [10,11]. Moreover, satisfactory results have been achieved in the molecular analysis of complex agronomic traits in rice [12]. Some QTLs related to effective panicle number in rice have gradually been identified. Xu et al. conducted a QTL analysis on yield-related traits using 292 recombinant inbred lines produced by TQ and LT. They detected a total of four QTLs affecting the EPN of rice on chromosomes 3, 4, 11, and 12 [13,14]. Zhang et al. constructed three sets of CSSL populations using PA64s, 9311, and Nihon [15]. After conducting whole genome resequencing, we obtained high-density physical maps. These maps allowed us to locate a QTL on chromosome 1 that controls EPN, with a contribution rate of 20% [15]. Wu et al. utilized the RIL population obtained from a cross between two indica rice varieties, ‘H359’ and ‘Acc85582’. Their findings revealed the presence of five QTLs responsible for controlling rice tillering traits, which were found on chromosomes 1, 3, and 5 [16]. Miyamoto et al. used an RIL population to locate four QTLs related to tillering, which were found on chromosomes 2, 5, 6, and 8, respectively [17]. In recent years, a large number of important agronomic trait-related genes in rice have been cloned, such as *MOC1*, *Ghd7*, *GS3*, *GW2*, etc. [12,18]. Li et al. discovered the single tiller gene *MOC1* using mutant materials. This gene is the first to be identified as controlling rice tillering and is located on chromosome 6 [19]. *TAD1* is a tillering and dwarf gene that encodes the coactivator of the anaphase-promoting complex/cyclosome (APC/C), a multi-subunit E3 ligase. It is located on chromosome 3 and is responsible for promoting the expression of the *OSH1* gene, thus maintaining the pre-meristem region and facilitating the formation of axillary meristem [20]. Although some genes related to the effective number of panicles in rice have been discovered, there are not many genes that can be applied to rice production and truly improve rice yield. Therefore, we still need to further explore more effective EPN-related genes.

In this study, rice varieties Dongfu 114 and Longyang 11 were used as parents to create an F_5_ population. Two extreme mixed pools were then constructed based on the analysis of effective panicles within the population. The main QTL interval of rice was obtained using the QTL-seq method, and the candidate genes associated with effective panicles were identified through haplotype analysis. The aims of this study were to explore new major loci in rice, provide valuable genetic resources for rice breeding, and have significant implications for studying rice tillering mechanisms and improving yield.

## 2. Results

### 2.1. Phenotypic Analysis and Evaluation of EPN

In this experiment, three replicates were set for the rice materials used, and of the two parental varieties, the EPN of ‘DF114’ plants was more than that of ‘LY11’ plants, and the EPN of the 309 F_5_ progenies (Figure 1) ranged from 11.98 to 18.25, with the 30 least and most progenies assigned to the L-pool and M-pool, respectively, for DNA resequencing. In addition, the skewness and kurtosis associated with EPN in the F_5_ population were 0.506 and 0.783, respectively. The average value of the F_5_ generation population was 14.32 (Appendix A). These values were consistent with the characteristics of quantitative traits overall, indicating that the data were suitable for QTL analysis.

### 2.2. BSA-Seq Analysis

The mean coverage depth for the parents and the two pools was 50 ×, and comparison of the sequences to the ‘Nipponbare’ reference genome resulted in the identification of 867,195 SNPs and 143,441 indels, which were reduced to 247,881 SNPs and 40,409 indels by trimming and filtering (Appendix A). A total of 288,290 high-quality SNPs/indels that were homozygous in each parent and polymorphic between the parents were then selected for BSA-seq analysis. ∆(SNP-Index) values (Figure 2A), Euclidean distance (ED) values (Figure 2B), and Fisher’s exact test *p*-values (Figure 2C) were used to identify candidate EPN-related QTL regions (Appendix A). Three significant (*p* < 0.01) peaks in the ED distribution spanned 24.62–26.47 Mb, 11.82–17.80 Mb, and 22.81–23.23 Mb on chromosomes 7, 9, and 11, respectively. In contrast, the peaks in the ∆(SNP-Index) and Fisher’s exact test *p*-value distributions covered the entire interval on chromosomes 7 and 11. Furthermore, after selecting intersections for each significant region, we identified significant regions on three chromosomes. These regions had fragment sizes of 1.85 Mb (24.62–26.47 Mb) on *qEPN7*, 6.41 Mb (15.49–21.90 Mb) on *qEPN9*, and 0.42 Mb (22.81–23.23 Mb) on *qEPN11*. Therefore, *qEPN7*, *qEPN9*, and *qEPN11* were considered to be more significant targets for mining candidate EPN genes.

### 2.3. Putative Candidate Genes for Three QTL Intervals

The three QTL intervals obtained from the above analysis were further screened, resulting in a total of 489 SNPs/indels, of which 238 were distributed on *qEPN7*, 242 on *qEPN9*, and 9 on *qEPN11*. After screening non-synonymous genes for upstream, UTR5, and exonic, 9 genes were obtained on *qEPN7*, namely *Os07g0602700*, *Os07g0602800*, *Os07g0602900*, *Os07g0603100*, *Os07g0603200*, *Os07g0603300*, *Os07g0603500*, *Os07g0614000*, and *Os07g0617600*. Additionally, 25 genes were obtained on *qEPN9*, namely *Os09g0431500*, *Os09g0433600*, *Os09g0473300*, *Os09g0509800*, *0s9g0526500*, *Os09g0526700*, *Os09g0531600*, *Os09g0532200 Os09g0533300*, *Os09g0535200*, *Os09g0535500*, *Os09g0538700*, *Os09g0539400*, *Os09g0539500*, *Os09g0540300*, *Os09g0542100*, *Os09g0546400*, *Os09g0547800*, *Os09g0549400*, *Os09g0549500*, *Os09g0549600*, *Os09g0550400*, *Os09g0551400*, *Os09g0551500*, and *Os09g0551600*, for a total of 34 genes.

### 2.4. Enrichment Analysis of Candidate Genes

This experiment conducted a GO enrichment analysis on 34 candidate genes and obtained significantly different GO terms. Some GO terms with higher enrichment were selected and plotted as pathway statistical maps (Figure 3A). According to different functional annotations, these GO terms can be divided into three categories: biological processes, cell component, and molecular function. The significant aspects of biological processes include DNA replication initiation, cellular response to reactive nitrogen species, and cellular response to inorganic substance. The most significant categories of cell composition include the Pwp2p–containing subcomplex of 90S preribosome, replication fork protection complex, and U12–type spliceosomal complex. Among the categories of molecular function, glycerophosphodiester phosphodiesterase activity, amine-lyase activity, and DNA replication origin binding have higher significance. It can be seen that these candidate genes may affect the tillering formation process of rice through pathways involved in biological processes, cell component, and molecular functions. In addition, this experiment also conducted KEGG enrichment analysis on 34 candidate genes. According to Figure 3B, these genes are primarily involved in basal transcription factors, galactose metabolism, ribosome biogenesis in eukaryotes, metabolic pathways, and other pathways. Except for metabolic pathways, these genes are significantly enriched in other pathways.

### 2.5. Temporal Expression Pattern of EPN-Associated Genes

Many studies have found that genes related to tillering exhibit a unique temporal expression pattern throughout the entire growth period. From 20 days (DAT) to 48 days (DAT) after transplantation, their expression levels remained stable and high at 00:00 (R0) and 12:00 (R12) in the roots, but decreased after 48 days [21]. This pattern can explain the stable transition from tillering development to ear development from a genetic perspective. Therefore, in this study, we attempted to explore the temporal expression patterns of genes related to EPN. We utilized the RichXPro website to conduct expression clustering analysis on root expression data obtained at weekly intervals of 00:00 (R0) and 12:00 (R12) for 34 candidate genes over the course of the growth period. The results showed that there were 32 genes with expression data in RiceXPro, and eight genes in R12 had relatively high expression levels from 20DAT to 48DAT (Figure 4A). In R0, there were nine genes with high expression levels from 21DAT to 49DAT (Figure 4B). After conducting a thorough comparison and screening of the two graphs, we identified six candidate genes that exhibited high expression levels in both time periods. These genes were *Os07g0603300*, *Os09g0433600*, *Os09g0549400*, *Os09g0539500*, *Os09g0549500*, and *Os09g0551600*. This expression pattern is consistent with the dynamic change pattern of tiller number throughout the entire growth period. There is a rapid increase in tiller number starting at 21DAT and reaching its peak at 49DAT. After 49 DAT, the expression of genes related to EPN decreased. This decrease may be associated with the transition from vegetative growth to reproductive growth and ear development on tillers.

### 2.6. Analysis of Candidate Gene Haplotype by RFGB Database

Through haplotype analysis of six candidate genes on the RFGB website, we found 12 SNP mutations in the *Os07g0603300* gene, including three in the promoter regions and nine in the coding regions. In total, we obtained 24 haplotypes. The EPN difference between haplotypes was significant. The *Os09g0433600* gene had nine SNP mutations, six promoter regions, and three coding regions. A total of 11 haplotypes were obtained, with significant differences in EPN among them. The *Os09g0549500* gene had 13 SNP mutations, four promoter regions, and nine coding regions. A total of five haplotypes were obtained, with significant differences in EPN among them. The *Os09g0549400* gene had 18 SNP mutations, four promoter regions, and 14 coding regions. A total of nine haplotype were obtained, with significant differences in EPN among them. The *Os09g0551600* gene had nine SNP mutations, six promoter regions, and three coding regions. A total of 13 haplotype were obtained, with significant differences in EPN among them (Figure 5). At the same time, we did not find any SNP mutations in *Os09g0539500*. Therefore, we ultimately obtained five candidate genes: *Os07g0603300*, *Os09g0433600*, *Os09g0549400*, *Os09g0549500*, and *Os09g0551600*. 

## 3. Discussion

Rice is the staple food for approximately 50% of the world’s population [22], making the demand for rice yield extremely significant. As one of the key elements, the EPN greatly affects rice yield and serves as the primary indicator for rice breeding. Improving rice EPN from a genetic perspective is worth exploring in depth. At present, some genes related to tillering have been identified in rice. For example, Zhu et al. constructed a low spike number chromosome segment substitution line (C3074) by backcrossing Japan’s Qinghe and Guanglu’ai 4 as parents. Using 1429 recessive individuals derived from NIL-F2–3 populations obtained from C3074 and Guanglu’ai 4, we successfully mapped the main QTL*qPN1* that controls the EPN per plant in rice. The QTL was finely mapped to a 34.4 kb region on the long arm of rice chromosome 1, which contains six annotated genes [23]. Chen et al. used Zhenshan97 and Miyang46 as parents to construct a backcross population. By screening multiple generations of self-crossing materials from the BC2 population, researchers constructed a fine mapping population with 18 target intervals that were separated and had a highly consistent genetic background. The micro effect QTL*qHd1*, which controls both heading date and panicle number, was finely mapped to a 950 kb region between RM12102 and RM12108 at the end of the long arm of rice chromosome 1. This locus has a significant effect on panicle number per plant, and it shows an additive effect of increasing efficiency on both heading date and yield traits [24]. However, continuous exploration of new EPN-related genes is crucial for improving rice yield. So far, few QTLs related to EPN have been identified. One reason is that the number of rice panicles is controlled by a polygenic system, with relatively low heritability and susceptibility to various environmental conditions [25]. Another reason is the absence of a genetically diverse population suitable for QTLs related to panicle numbers. In addition, there is often a negative correlation between EPN and the other two yield components [2]. Improving one gene factor often has a negative impact on other trait factors. High yield at the individual level of rice does not necessarily mean high yield at the population level. How to coordinate the balance between the source and sink of rice, and make optimal use of rice yield-related genes in rice breeding, is the crucial issue. This study selected two rice varieties, Dongfu 114 and Longyang 11, as parents, based on their significant differences in effective panicle number (EPN). The aim of the study was to identify excellent QTLs and candidate genes associated with higher EPN in rice. This will be achieved through methods such as BSA-seq analysis. The findings of this study will provide a theoretical foundation for future efforts to improve rice yield.

The BSA-seq method can rapidly localize gene QTLs. With the advancement of sequencing technology, the cost has significantly decreased. Compared to traditional linkage mapping, QTL-seq can improve work efficiency and provide high-density mutation sites. QTL-seq has been successfully applied to many plants, including cucumbers [26], soybeans [27], rice [28], and tomatoes [29]. With the recent advancements in high-throughput sequencing, high-resolution mass spectrometry analysis, and information processing technology, BSA-seq has become more mature. Its accuracy and cost have significantly improved [30]. The combination of traditional maps and BSA-seq can effectively and quickly narrow down the main QTL intervals [9,31,32]. For example, Guo et al. identified candidate genes for controlling drought tolerance in rice grown in cold regions during fertilization using BSA-seq and RNA-seq [33]. Zhao et al. identified a new site, *qGL3.5*, that regulates rice grain length using BSA-seq [34]. Liang et al. combined a large number of isolation analyses with BSA-seq to identify a new type of *pi21* haplotype, thus endowing rice with innate resistance to rice blast [35]. The main advantage of the BSA-seq method is that it utilizes DNA mixed gene pool sequencing. It uses the genotype of the control parent as a reference to calculate the SNP index in the extreme pool of offspring. This reduces the cost of DNA extraction and allows for the direct utilization of polymorphic SNPs between two parents for localization, without the need for new markers [36]. F2, RIL, and DH populations can all serve as target populations for QTL-seq. While screening candidate regions, it is possible to perform functional analysis of candidate genes within those regions, which may lead to precise localization in a single step [9,37]. In this study, three bioinformatics analysis methods were used to map the QTL region at the 99% significance level. Using the BSA-seq strategy, QTLs related to effective panicle number were located on chromosomes 7, 9, and 11. The identified QTLs were found in three regions containing annotated genes. In general, phenotypic variation is caused by non-synonymous mutations in the gene coding region of genes. Therefore, in this study, we first considered which regions of the mapped QTL contained non-synonymous significant SNPs, and the variation in the promoter or CDS region was likely to be the key to gene expression regulation. In order to screen potential candidate genes in the interval, we combined QTL-seq with time expression pattern and haplotype analysis. This approach reduced the number of candidate genes in the interval defined by QTL-seq from 34 to 5.

In the past decade, genotype datasets for many rice varieties have been published and used to identify several loci related to important agronomic traits [38,39]. Some large-scale rice collections with sequence and phenotype data provide valuable materials and knowledge for rice research and breeding projects [38,40,41,42]. We found that *Os07g0603300* and *Os09g0549500*, two out of the five candidate genes, have been discovered and cloned by previous researchers. *Os07g0603300* is *GL7*, a major QTL that controls grain length and width on rice chromosome 7. It encodes a homologous protein to Arabidopsis LONGIFOLIA protein and regulates the longitudinal elongation of cells. The 17.1 kb tandem repeat at the *GL7* locus causes the upregulation of *GL7* expression level and downregulation of the expression of negative factors adjacent to GL7, thereby increasing the grain length and improving the appearance quality of rice [43]. *Os09g0549500*, also known as *OsU11*/*U12-31K*, is necessary for normal plant development. Artificial microRNAs (amiRNA) knock out *AtU1*/*U1231k* mutant plants, resulting in delayed main stem growth. However, rice *Os31K* has the ability to restore the wild-type phenotype to Arabidopsis plants with amiR1-4 [44]. Nowadays, with the continuous development of bioinformatics technology and the gradual reduction in sequencing costs, many studies on rice tillering traits are not limited only to the initial positioning research, but also pursuing more detailed, accurate, and in-depth explorations. Although some progress has been made in studying rice tillering ability, the available information is still quite limited. Therefore, on one hand, it is necessary to continue exploring more reliable gene information in the rice genome. On the other hand, it is important to quickly locate and clone functional genes using mutant materials. Only by accelerating the exploration of the rice functional genome can we effectively guarantee future gene cloning. Due to time constraints and various factors, this study did not conduct further research on the selected candidate genes. The next step could be to prioritize the remaining three candidate genes for in-depth research. This would involve exploring the cloning, genetic transformation, and regulation of rice tillering mechanisms associated with these genes. It is important to validate these findings in order to provide a more comprehensive understanding of the genetic mechanism of rice tillering.

## 4. Materials and Methods

### 4.1. Plant Materials and Construction of Segregating Pools

The japonica varieties ‘Dongfu 114’ (‘DF114’) and ‘Longyang 11’ (‘LY11’) were obtained from Northeast Agriculture University (Harbin, China) and used as the female and male parents, respectively, to create an F_5_ population of 309 individuals. In the spring of 2021, ‘DF114’ (*n* = 48), ‘LY11’ (*n* = 48), and F_5_ (*n* = 309) individuals were planted in four rows under natural conditions in paddy fields at Acheng Experimental Station (Harbin, Heilongjiang Province, China), and 5 plants from the center of each plot were selected for evaluation of EPN. Based on the analysis of the EPN of 309 F_5_ individual plants, 30 plants with the minimum number of panicles and 30 plants with the maximum number of panicles were selected as the extreme few panicle pool (L-pool) and extreme many panicle pool (M-pool), respectively.

### 4.2. Phenotyping Analysis Genotyping Data and SNP Filtering

DNA samples from Dongfu 114, Longyang 11, and two pools with extreme mixing were selected and submitted to Guangzhou Kideo Biotechnology Co., Ltd. (Guangzhou, China) for QTL-seq analysis. First, DNA samples were collected. The integrity of the DNA was assessed using agarose gel electrophoresis. The DNA concentration was measured using Nanodrop, and the accurate quantification of DNA concentration was performed using Qubit. The IlluminaHiSeq platform was used for sequencing. The average coverage depth of the parents and two mixing tanks was 50×. After sequencing using the IlluminaHiSeq platform, the original sequencing data, RawData, was obtained. Quality checks and filtering were performed to obtain CleanReads. BWA-backtrack software was used to compare CleanReads with the Nipponbare reference genome. Picard 2 software was utilized to remove duplicate readings. GATK 4 software was employed to detect and filter SNPs. Finally, SNP sites between the sample and the reference genome were identified. Association analysis was performed using ∆(SNP-Index) [45], ED [46], and two-tailed Fisher’s exact test values [47]. The final QTL interval was determined by considering the overlapping interval of the three methods.

The SNP index is a method of searching for QTL loci by looking for differences in genotype frequencies between mixed pools. If the genotype changes of a certain locus are not related to phenotype, the proportion of alleles in the two mixed pools should be roughly equal, and the SNP index is close to the theoretical separation ratio. When Δ (SNP index) is close to 0, it indicates that there is little correlation between the marker SNP and the trait. Conversely, when Δ (SNP index) is closer to 1, it suggests a stronger correlation between the marker SNP and the trait, making the chromosome region a potential candidate region for QTL.

ED is one of the methods that uses sequencing data to identify significant difference markers between mixed pools and evaluate the regions associated with traits. The ED value of the non-target site should tend toward 0. The larger the ED value, the greater the difference between the two mixed pools of the marker.

The two-tailed Fisher’s exact test is based on the hypergeometric distribution. It is used to test the allele depth ratio in two mixed pools and determine if there is a significant difference. This significance is represented by a *p*-value. The smaller the *p*-value, the higher the significance, indicating a greater possibility of a difference in the allele ratio between the two mixed pools.

### 4.3. GO and KEGG Enrichment Analysis of Candidate Genes

The candidate genes within the QTL intervals obtained from QTL-seq analysis were annotated using the online software Ensembl (http://www.ensembl.org/index.html/2023/04/25/) with Nipponbare as the reference genome. Firstly, we mapped genes to each term in the GO database (http://www.geneontology.org/2023/5/10/). and calculated the number of genes in each term. Then, we applied hypergeometric tests to identify GO entries that were significantly enriched in genes compared to the background of the entire genome. We then utilized GO analysis to classify genes based on their cell component, molecular function, and biology. KEGG is the primary public database related to Pathways. Pathway significance enrichment analysis utilizes KEGG Pathway as a unit and employs hypergeometric tests to identify pathways that are significantly enriched in genes compared to the background of the entire genome. Significant enrichment through Pathway can determine the main biochemical, metabolic, and signal transduction pathways involved in genes. We analyzed the metabolic pathways of these annotated genes using the KEGG website (https:///www.kegg.jp/2023/5/10/).

### 4.4. Temporal Expression Pattern and Haplotype Analysis

Through the RichXPro website, expression clustering analysis was conducted on the root expression data of candidate genes collected at weekly intervals of 00:00 (R0) and 12:00 (R12) throughout the growth period. The purpose was to screen for genes with higher expression levels. Through the Haplotype Analysis module (HaplotypeAnalysis) of the RFGB database (https://www.rmbreeding.cn/2023/5/25/), we analyzed the mutation information of the candidate gene promoter and the CDS region, obtained each haplotype of the candidate gene and further screened it. Then, we used the PN phenotype database to analyze the difference in the number of spikes among haplotypes.

### 4.5. Statistical Analysis

Differences between parent and progeny EPN were analyzed using IBM SPSS Statistics 26. The average, standard deviation, skewness, and kurtosis of the EPN statistics were calculated. Graphs were created using Adobe Photoshop CC 2019 and Origin 2021.

## 5. Conclusions

This study utilized the japonica rice varieties Dongfu 114 and Longyang 11 to establish an F_5_ population comprising 309 individual plants. By identifying the EPN in the population, 30 individual plants with extreme traits were selected. QTL-seq analysis was then conducted on the parents, as well as on pools of plants with multiple panicles and plants with few panicles. The ∆ (SNP Index) algorithm, ED algorithm, and Fisher exact test values collectively identified the primary effective QTL intervals for EPN on chromosomes 7, 9, and 11 as 1.85 Mb (24.62–26.47 Mb), 6.41 Mb (15.49–21.90 Mb), and 0.42 Mb (22.81–23.23 Mb), respectively. Afterwards, temporal expression pattern analysis of EPN was performed on 34 genes to identify six candidate genes with higher expression levels. Then, haplotype analysis was performed on these six genes using a 3K database to screen out genes with insignificant haplotypes. This process resulted in the identification of five high-quality candidate genes. Although further work is needed to elucidate the mechanisms of action of these genes, this study provides resources for breeding programs aimed at improving the EPN.

## Figures and Tables

**Figure 1 ijms-24-14838-f001:**
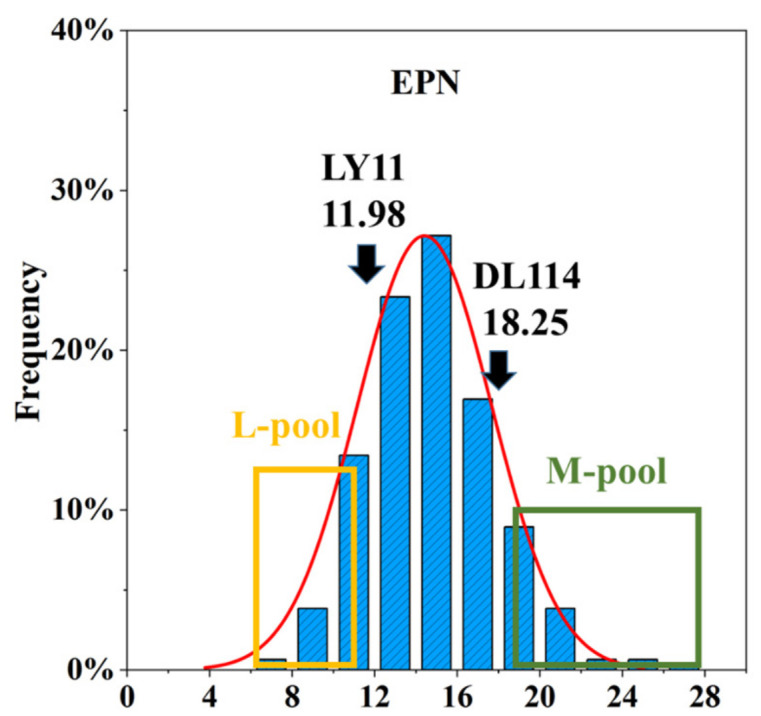
Probability distribution of EPN of 309 rice plants in F_5_ population.

**Figure 2 ijms-24-14838-f002:**
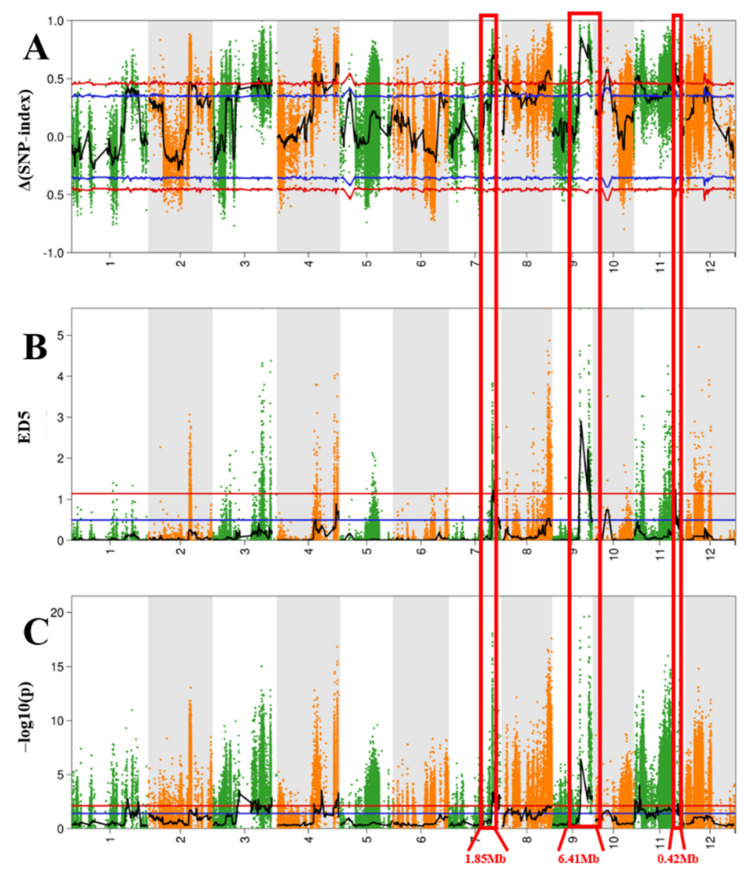
Quantitative trait locus (QTL) analysis of rice EPN at maturity using three QTL-seq methods. (**A**) Manhattan plot showing the distribution of Δ(SNP-index) on chromosomes. (**B**) Manhattan plot showing the distribution of Euclidean distance (ED5) on chromosomes. (**C**) Manhattan plot showing the distribution of log-transformed Fisher’s exact test *p*-value distribution, –log10(p) on chromosomes. Blue and red lines represent 95 and 99% confidence intervals, respectively, and black lines represent mean values of the three algorithms, which were drawn using sliding window analysis. Numbers on the horizontal coordinates represent chromosome numbers.

**Figure 3 ijms-24-14838-f003:**
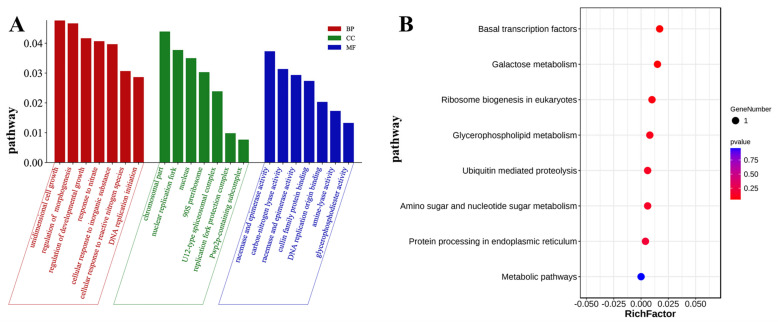
GO enrichment analysis and KEGG enrichment analysis for candidate genes. (**A**) GO analysis. (**B**) KEGG analysis.

**Figure 4 ijms-24-14838-f004:**
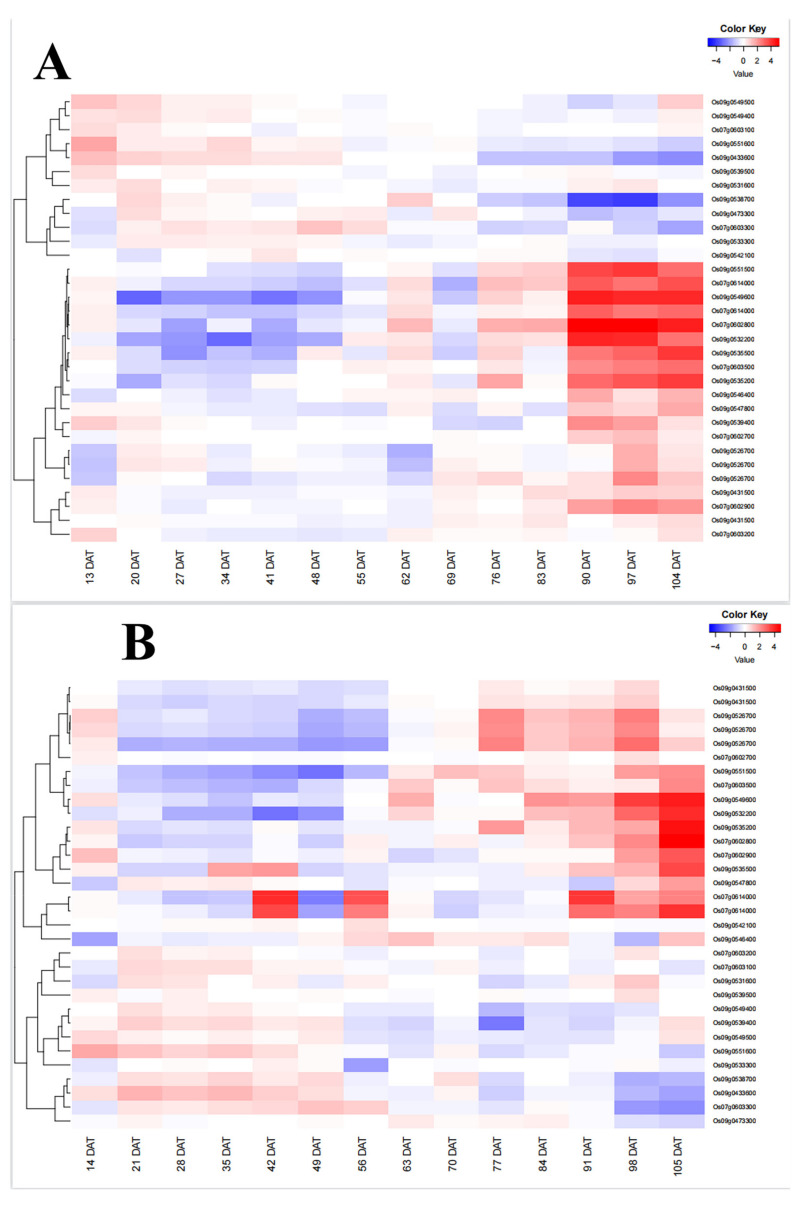
Temporal expression pattern of EPN-associated genes during the whole growth period. Expression data in root at 00:00 (**A**) and 12:00 (**B**) were downloaded from RiceXPro website. The heatmaps represented hierarchical clustering of relative expression levels of 27 candidate genes at different days after transplanting (DAT). The scale for relative expression levels (after normalization by z-score) is denoted by color bars, with red representing the high expression levels, white medium expression, and blue low expression.

**Figure 5 ijms-24-14838-f005:**
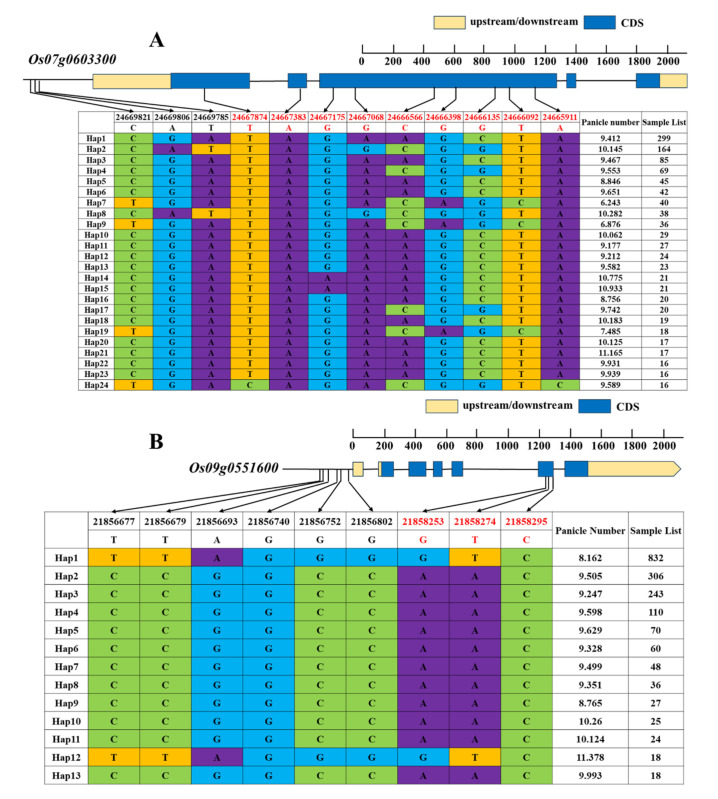
Haplotype analysis of candidate genes. (**A**) Haplotype analysis of *Os07g0603300*. (**B**) Haplotype analysis of *Os09g0551600*. (**C**) Haplotype analysis of Os09g0433600. (**D**) Haplotype analysis of *Os09g0549500*. (**E**) Haplotype analysis of *Os09g0549400*.

## Data Availability

No new data was created.

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
