# Peer review of "BSA-Seq for the Identification of Major Genes for EPN in Rice"

_ijms, 2023, doi:10.3390/ijms241914838_

Round 1

Reviewer 1 Report

I like this study as the authors used BSA-seq for the Identification of Major Genes for Effective Panicle NumberEPN in Rice. to study the whole experimental work.

However, the current MS needs minor modifications before acceptance

Some sentences and para need proper revision for more clarity as in the abstract and discussion. Overall significance is missing.

Figure quality needs to improve along with significance analaysis.

Grmmatical improvement throughout the manuscript is required.

Reviewer 2 Report

The manuscript by Shen et al  entitled “BSA-seq for the identification of major genes for EPN in rice” investigated genetic control for effective panicle number in rice. They identified QTLs using BSA combined with NGS, mined for candidate genes under 3 QTL regions and narrowed down to 5 genes.

Comments:

1.      Method section is poorly described. Specially, authors mentioned that they used 3 different QTL-seq methods. BSA-seq, Euclidean distance and Fisher exact test. None of these methods are explained anywhere. Well detailed description is needed.

2.      There are several other signals are present on chromosome in almost all chromosomes. What is the rational of using only three QTL on chromosome 7 ,9 and 11.

3.      All QTLs identified in this study should be numbered and consistent with the existing nomenclature

4.      Figure 5 of Haplotype analysis is not readable. Authors claim haplotype are associated with phenotype. Need readable data and statistical analysis. Also are those phenotypes collected by authors?

5.       All figure need better resolution and detailed explanation in legend.

Thanks

Reviewer 3 Report

The manuscript submitted for review raises interesting issues, however, a few comments to the authors:

1.      Not very readable chart 3,5 hard to read x axis. I think graph 5, which is divided into A,B,C,D,E, should be done vertically rather than in two columns to make the results more readable

2.      Spacing between words, e.g. line 10, 11- this is an example of the lines of an article- note applies to the whole manuscript

3.      Line 20 - between sentences - "genes with higher expression levels; Using a 3K database to perform haplotype" perhaps a full stop?

4.      Line 12 Currently, the main QTL - it would be good to expand on this acronym

5.      To quote the authors of eq. ".....mainly determined by the yield composition factors[2]." - after the text there should be a space and then a parenthesis- it is differently recorded in whole manuscript

6.      Introduction - The EPN per plant in rice is easily affected by the number of tillers, and the number of tillers in rice is also constrained by many factors, such as genetic background and fertilizer, which can affect the number of tillers in rice.- hmm it would be good to improve this sentence, a sentence difficult to understand, sentence too long, entangled,

7.      A total of 4 QTLs affecting the EPN of rice were detected on chromosomes 3, 4, 11, and 12[13,14]ï¼›probably a full stop at the end?- Note applies to the entire manuscript

8.      Ln. 76 - Zhang et al. […..]??? constructed three sets of CSSLs populations using PA64s, 9311, Ln 260, 261 ‘….regions during pregnancy using BSA-seq and RNA-seq[33]; Zhao et al. identified a new site qGL3.5 that regulates rice grain length using BSA-seq[34]; Liang et al. combined a large…’? -  no number to the list of journals - note to all entries throughout the manuscript

9.       Can diagram 4, be of better quality?

10.   Please review all full stops, commas - note to entire manuscript. Lack of appropriate sentence endings, causes difficulty in reading the manuscript

11.    Line 124 '.....distributions completely covered the interval on Chromosome 7 and 11.Furthermore,after..... are there any spaces in this sentence towards the end?

12.   Figure 5. Haplotype analysis for candidate genes(A) Haplotype analysis of Os07g0603300.(B) Haplotype analysis of Os09g0551600.(C) Haplotype analysis of Os09g0433600.(D) Haplotype analysis of Os09g0549500.(E) Haplotype analysis of Os09g0549400- caption of Figure 5- no spaces, which is confusing

13.   4.5 Statistical Analysis - Differences between parent and progeny EPN were detected using SPSS 26.0,and calculate the average, standard deviation, skewness and kurtosis of the EPN statistics.Graphs were drawn using Adobe Photoshop CC 2019 and Origin 2021.What is this SPSS 26.0? - There is no information about the type of test, and in the text it is what type of test was used

Round 2

Reviewer 2 Report

Authors addressed my comments in revised version and improved accordingly.

Reviewer 3 Report

The authors have made the indicated corrections. They have also addressed all comments. In my opinion, the article can be forwarded for publication.